# A rat epigenetic clock recapitulates phenotypic aging and co-localizes with heterochromatin

Morgan Levine[1,2]*, Ross A McDevitt[3], Margarita Meer[1], Kathy Perdue[3], Andrea Di Francesco[4,5‡§], Theresa Meade[3], Colin Farrell[6], Kyra Thrush[2], Meng Wang[2], Christopher Dunn[7], Matteo Pellegrini[8], Rafael de Cabo[4†]*, Luigi Ferrucci[4†]*

[1]Department of Pathology, Yale University School of Medicine, New Haven, United States; [2]Program in Computational Biology and Bioinformatics, Yale University, New Haven, United States; [3]Comparative Medicine Section, Biomedical Research Center, National Institute on Aging, National Institutes of Health, Baltimore, United States; [4]Translational Gerontology Branch, Biomedical Research Center, National Institute on Aging, National Institutes of Health, Baltimore, United States; [5]Calico Life Sciences, South San Francisco, United States; [6]Department of Human Genetics, University of California, Los Angeles, Los Angeles, United States; [7]Laboratory of Molecular Biology and Immunology, Flow Core Unit, Biomedical Research Center, National Institute on Aging, National Institutes of Health, Baltimore, United States; [8]Molecular Biology Institute and Departments of Energy Laboratory of Structural Biology and Molecular Medicine, and Chemistry and Biochemistry, University of California, Los Angeles, Los Angeles, United States

*For correspondence:
morgan.levine@yale.edu (ML);
decabora@grc.nia.nih.gov (RC);
FerrucciLu@grc.nia.nih.gov (LF)

†These authors contributed equally to this work

Present address: ‡Office of Research Oversight, Veterans Health Administration, US Department of Veterans Affairs, Washington DC, United States; §Calico Life Sciences, San Francisco, United States

Competing interests: The authors declare that no competing interests exist.

**Abstract** Robust biomarkers of aging have been developed from DNA methylation in humans and more recently, in mice. This study aimed to generate a novel epigenetic clock in rats—a model with unique physical, physiological, and biochemical advantages—by incorporating behavioral data, unsupervised machine learning, and network analysis to identify epigenetic signals that not only track with age, but also relates to phenotypic aging. Reduced representation bisulfite sequencing (RRBS) data was used to train an epigenetic age (DNAmAge) measure in Fischer 344 CDF (F344) rats. This measure correlated with age at (r = 0.93) in an independent sample, and related to physical functioning (p=5.9e-3), after adjusting for age and cell counts. DNAmAge was also found to correlate with age in male C57BL/6 mice (r = 0.79), and was decreased in response to caloric restriction. Our signatures driven by CpGs in intergenic regions that showed substantial overlap with H3K9me3, H3K27me3, and E2F1 transcriptional factor binding.

## Introduction

Since first appearing in the literature in 2011 (*Bocklandt et al., 2011*), 'epigenetic clocks' have emerged as some of the most promising potential biomarkers of aging (*Horvath and Raj, 2018*). Epigenetic clocks are traditionally measured by combining information on DNA methylation (DNAm) levels at hundreds of cytosine-guanine dinucleotides (CpGs) to produce age predictions. These estimates have been shown to strongly correlate with observed age (often r > 0.7) (*Horvath, 2013*; *Hannum et al., 2013*; *Meer et al., 2018*; *Petkovich et al., 2017*; *Thompson et al., 2018*). More importantly, the divergence in the predicted age relative to the observed age has been shown to reflect differential mortality and/or disease risk/prevalence (*Levine et al., 2015a*; *Levine et al.,*

*2015b*; *Levine et al., 2016*; *Levine et al., 2018*; *Ambatipudi et al., 2017*; *Horvath et al., 2014*; *Horvath et al., 2015*; *Lu, 2019*)—suggesting it captures differences in biological rather than chronological aging. Furthermore, in mice, evidence is beginning to emerge to show that genetic or behavioral interventions known to influence aging produce differences in the epigenetic age of birth cohorts (*Meer et al., 2018*; *Petkovich et al., 2017*; *Thompson et al., 2018*; *Sziraki et al., 2018*)—leading to the growing enthusiasm surrounding the application of epigenetic clocks as powerful biomarkers in aging research.

The quest to identify measures that can assess biological age is currently a major goal in Geroscience research (*Justice et al., 2018*; *Kennedy et al., 2014*; *Sierra, 2016*). With the growing list of potential therapeutics to target aging, there is an immediate need to establish gold-standard measures for efficacy testing. Accurate biomarkers of aging will also enable better prognostic evaluation; they may inform treatment, study inclusion, and personal decisions; and perhaps most importantly, they will help uncover the underlying mechanisms driving biological aging. However, in order for a biomarker to be valuable in these endeavors, we must distinguish between changes that simply track with chronological time from those that relate to biological function (both entropic damage and/or compensatory mechanisms).

While up to this point, the vast majority of epigenetic clock studies have been conducted using human cohorts, our ability to elucidate the underlying biology of epigenetic aging will require mechanistic studies using mammalian animal models. Epigenetic clocks based on blood and multiple tissues have been developed for mouse models (*Meer et al., 2018*; *Petkovich et al., 2017*; *Thompson et al., 2018*; *Stubbs et al., 2017*); however, because of their small size, drawing sufficient blood volumes for epigenetic analysis require terminal bleeds. Thus, the use of mice presents a problem when it comes to tracking epigenetic aging longitudinally or using it as a prognostic indicator. Conversely, rats share many of the same advantages as mice, yet also are approximately 10 times larger in mass, enabling the safe collection of substantially more blood at a given point in time without undue harm to the animal (*Ellenbroek and Youn, 2016*). The ability to safely collect a larger quantity of blood will allow researchers to track animals longitudinally and relate molecular changes to changes in phenotypic characteristics—including in vivo brain imaging or individual differences in age-related cognitive decline, for which rats may be better models than mice (*Ellenbroek and Youn, 2016*). Allowing for multiple assays from the same samples will enable researchers to draw links between various hallmarks of aging.

The ability to link epigenetics to other phenotypic data is of critical importance in developing biomarkers of aging. Traditionally, correlations with chronological age have been used to assess validity of aging biomarkers (*Horvath and Raj, 2018*; *Hannum et al., 2013*; *Meer et al., 2018*; *Petkovich et al., 2017*; *Thompson et al., 2018*; *Horvath, 2013*; *Stubbs et al., 2017*). However, evidence is mounting that this may not be the best approach. Indeed, the epigenetic clocks that appear most informative for predicting future health and wellness are not necessarily the strongest age predictors (*Horvath and Raj, 2018*; *Levine et al., 2018*). This is due to the fact that a number of molecular and physiological traits change over the lifespan, but the degree of change does not necessarily reflect their importance in the biological aging processes manifesting as death and disease. Therefore, the only way to distinguish changes that track chronological time versus those that track biological aging is to validate potential biomarkers using variables other than age.

The aim of this study was to measure DNAm using blood samples from a large, multi-age cohort of Fischer 344 CDF (F344) rats (n = 134) and associate DNAm patterns with chronological age along with behavioral and cellular changes. In doing so, we identified key epigenetic age changes and developed a novel rat epigenetic clock, which we show tracks age and functioning in rats. We also demonstrate that this clock can be used to track aging in mice and is responsive to interventions such as caloric restriction and cellular reprogramming.

## Results

### Age differences in DNAm

Male F344 rats were acquired from the NIA Aged Rodent Colony. The samples were evenly distributed over a large age range, spanning 1 to 27 months, with six rats in each age group (n = 162). Within three weeks of arrival, rats were weighed, blood was drawn for FACS and methylation

sequencing, and behavioral testing was performed. Our analytic sample included 134 rats. Excluded samples represent animals that died or were euthanized prior to blood draw, those that did not pass quality control for bisulfite sequencing, and/or those without complete FACS, rotarod, or open field data.

Whole blood from retro-orbital sampling was used to perform reduced representation bisulfite sequencing (RRBS) and imputation was conducted via a k-nearest neighbor (kNN) sliding window, yielding information on DNAm levels at 5,505,909 cytosines across the genome for all animals. To train an age predictor, samples were split into training (n = 102) and test sets (n = 32) based on age to ensure equal age distributions. For instance, rats ages 2 m, 6 m, 10 m, 14 m, 18 m, 22 m, and 26 m were selected for the test set (n = 32), while all others were included in the training set (1 m, 3 m, 4 m, 5 m, 7 m, 8 m, 11 m, 12 m, 13 m, 15 m, 16 m, 17 m, 19 m, 20 m, 21 m, 23 m, 24 m, 25 m, 27 m). By applying elastic net regression to the training set, we built an age predictor that selected 68 of the 5.5 million sites. The age predictor based on the 68 CpGs, which we call DNAmAge, exhibited a correlation of r = 0.90 with observed age in the validation sample (*Figure 1A*). As with many of the other human and rodent clocks, the relationship between DNAmAge and age exhibits nonlinear properties, suggesting a sigmoidal relationship between the two variables.

We hypothesize that the aging trends exhibited across the methylome are highly redundant, and as a result, dimensionality can be greatly reduced without compromising the underlying signal being captured. To test this, we performed principal component analysis (PCA) using the training sample to identify orthogonal signals and test for associations with age. Results suggest that PC1 only captures 4.7% of the variance in our data, and that jointly, the first 10 PCs capture 28% of the variance (*Figure 1B*). Nevertheless, PC1 was strongly correlated with age (r = 0.93) in our independent test sample (*Figure 1C*). Furthermore, when we applied these PCs (in contrast to CpGs) to train an age predictor, based on 10-fold cross-validation in the training sample, our model suggested that age prediction was not improved by including PCs beyond PC1 alone (*Figure 1—figure supplement 1*). Because eigenvalues for PC1 are in standardized units, we converted these to units of months to generate an interpretable estimate of DNAmAge. This was done using the beta coefficient and intercept values from a linear regression of chronological age regressed on PC1. To convert to units of months, PC1 eigenvalues were multiplied by the beta coefficient (0.405) and summed with the intercept value (13.966). This measure of DNAmAge is perfectly correlated (r = 1) with the original PC1 eigenvalues, thus, this rescaling does not alter our results. Rescaling using the coefficient 0.405 and constant 13.966 was applied or all subsequent validation in independent test samples.

## DNAm associations with FACS and phenotypic variables

To explore the association between DNAm and aging-related functional variables, we tested the chronological age-adjusted associations between the DNAmAge measures and performance in rotarod and open-field behavioral tests of locomotor function. Data was available for eight summarized open-field test variables and two rotarod variables (max time and mean time). Additionally, we also tested for confounding variables heterogeneity in blood cell types using Fluorescence Activated Cell Sorter (FACS) comprising information on 39 variables related to white blood cell composition. As with the DNAm data, given the redundancy in these measures, we ran two PCAs using the training sample—one to reduce the dimensionality of the FACS data and one to reduce dimensionality of the behavioral phenotype data (open field and rotarod). When examining the age correlations of the first three PCs, we found that PC1 for both types of variables are strongly associated with age in the validation sample (*Figure 1—figure supplement 2*). PC1 based on FACS data shows a correlation of r = 0.91 with age, and the PC1 based on phenotype data (open field and rotarod) correlates with age at r = 0.63. As shown in *Supplementary file 1* Table S1, PC1 for phenotype data is most strongly related to the open-field variables, except for time spent in the center zone (an anxiety-sensitive measure) and duration of stereotypic movements, including grooming and sniffing. When examining association between age and individual phenotypic variables or FACS variables, we observed the same strong trends (*Figure 1—figure supplement 3* and *Figure 1—figure supplement 4*).

Using multivariate linear regression models, we tested whether age-adjusted DNAmAge was associated with phenotypic variables and/or confounded by FACS (*Table 1*). In model 1, which included DNAmAge regressed on chronological age alone, as expected we observed that DNAmAge is strongly related to chronological age—similar to the correlation results. Results for Model

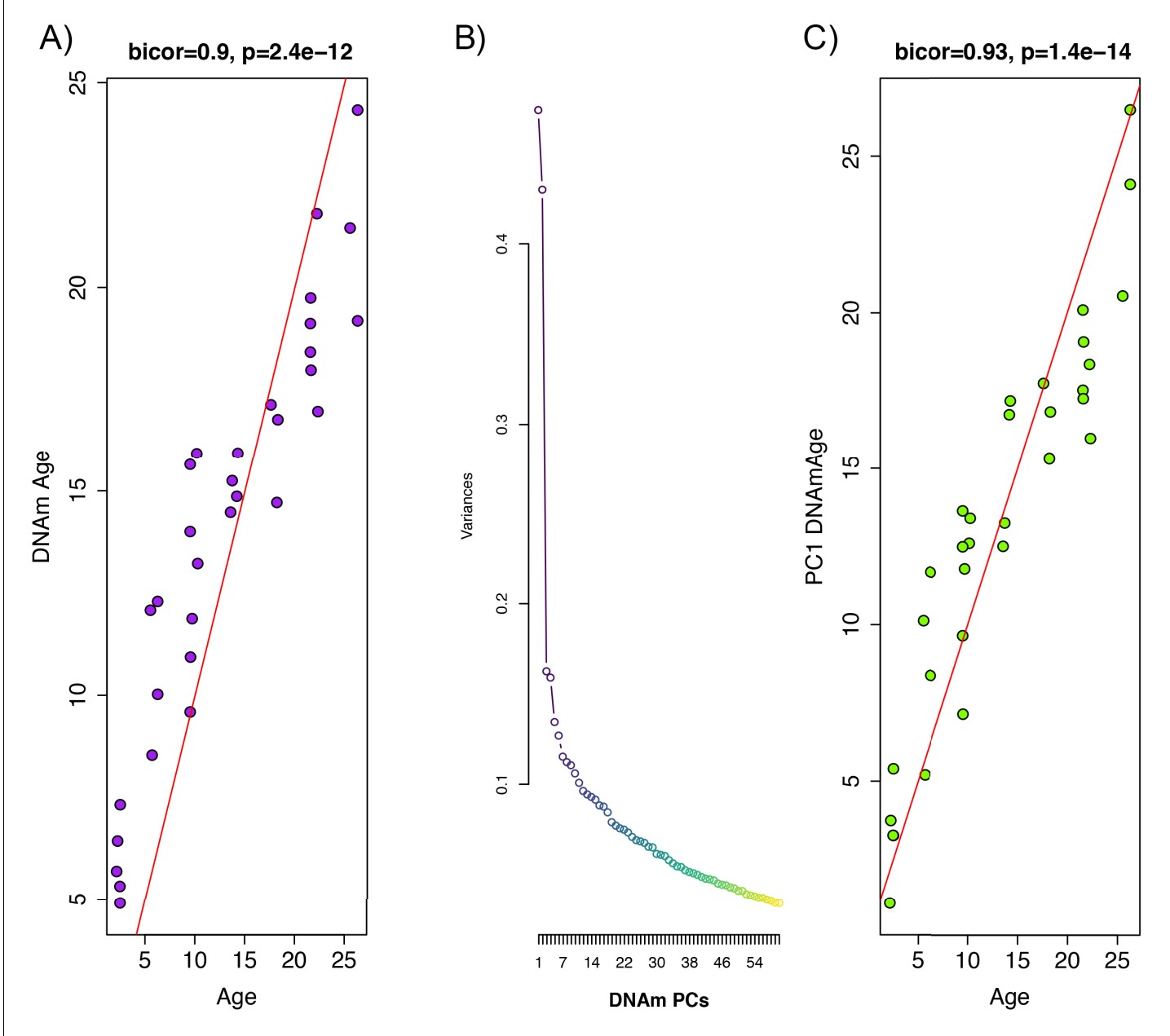

**Figure 1.** Principal component analysis (PCA)-based construction of DNAmAge in rats. (**A**) Supervised approach: A sample of male F344 rats, ages 1 to 27 months, were split into training (n = 102) and test sets (n = 32). Using DNAm levels measured at 5,505,909 CpGs across the genome as input, we applied elastic net penalized regression to train a predictor of age. Based on training data, 68 CpGs were selected for the DNAmAge measure. The plot shows the biweight midcorrelation between age (x-axis) and DNAmAge (y-axis) in the test sample. (**B**) Screeplot of variance explained for PCs1-60. A large amount of the variance is captured by PCs 1 and 2, with an elbow forming around PC7, suggesting that the amount of variance explained drops-off significantly after the first six PCs. (**C**) Unsupervised approach: we converted PC1 (estimated based on PCA in the training sample) to units of months by regressing age on PC1 and then multiplying the coefficient by PC1 and adding the constant in the test sample. We then tested the biweight midcorrelation between PC1 DNAmAge and chronological age.

The online version of this article includes the following figure supplement(s) for figure 1:

**Figure supplement 1.** Tenfold cross-validation of prediction of chronological age using eigenvalues from PCA of DNAm.

**Figure supplement 2.** Age associations for PCs1-3 from FACS and phenotype data.

**Figure supplement 3.** Associations between chronological age and phenotypic data (open field and rotarod).

**Figure supplement 4.** Associations between chronological age and FACS data.

**Table 1.** Pheno PC1 and FACS PC1 are standardized variables, such that they have a standard deviation of 1.

Given that the DNAmAge variable is in units of (months), the beta coefficients represent the increase in DNAmAge (in units of months) associated with every 1 s.d. increase in either PC1 FACS or PC1 Pheno.

|  | Beta coefficient (P-value) | | |
|---|---|---|---|
|  | **Model 1** | **Model 2** | **Model 3** |
| DNAmAge (PC1) | | | |
| Age | 0.75 (6.0e-15) | 0.63 (1.27e-11) | 0.42 (4.4e-4) |
| Pheno PC1 | – | 1.46 (3.4e-3) | 1.29 (5.9e-3) |
| FACS PC1 |  | – | 1.91 (2.7e-2) |

2, which included the addition of the phenotypic PC1 (standardized to have s.d. = 1), showed a significant association between DNAmAge and the physical functioning after controlling for age in the model. This suggests that among rats of the same chronological age, every one standard deviation in worsening locomotor function is associated with a 0.43 month increase in DNAmAge. Moreover, the addition of the phenotypic variable somewhat attenuated the chronological age effect, suggesting that part of the age-associated increase in DNAmAge is explained by declines in physical functioning (16%). In Model 3, we included PC1 from the FACS data (standardized to have s.d. = 1) as an additional covariate. In doing so, we observed further attenuation of the age association and a small attenuation of the phenotypic association, suggesting that cell composition may account for a small proportion of the associations between DNAmAge and either age and/or physical functioning. Nevertheless, the associations with phenotypic PC1 remained significant, suggesting that among animals of the same chronological age, age-related functional decline is related to epigenetic aging independent of cell composition. This was further supported by an expanded model in which the association between phenotypic PC1 and DNAmAge remains (β = 1.7, p=6.3e-3), even after adjustment for the first 10 PCs from the FACS PCA (results not shown).

## Validation in C57BL/6 mice

To add additional out-of-sample validation, we used publicly available RRBS data from mouse models. However, given the sparsity in RRBS coverage, after mapping genomic coordinates from rat (rn6) to mouse (mm10), only 3,625 CpGs were available across all samples. Given this limitation, we re-calculated PC1 using only CpGs in common between the two species. When re-evaluating the performance in the rat data, the new DNAmAge measure from the overlapped CpGs was able to equivalently recapture the age signal, exhibiting a correlation of r=0.92 (*Figure 2A*) in the test data from rats. This is despite the fact that we started with a much smaller number of parameters (~3,600 compared to 5.5 million). Furthermore, the association between epigenetic aging and phenotypic aging in rats remained, and if anything, was slightly more robust (*Supplementary file 1* Table S2).

To estimate DNAmAge in mice, we applied the PCA model that was calculated in the rat data using the 3625 overlapped CpGs to the overlapping mouse RRBS data. We found that DNAmAge explains 62% of the variance in age in the mouse validation sample (r=0.79 correlation). Interestingly, the mouse data showed high intercept and lower slope for DNAmAge as a function of age, suggesting that this DNAmAge measure is initially over-estimated in mouse, but that the rate of increase with age is slower than for rats. However, again, we found a non-linear sigmoidal association between DNAmAge and chronological age in both the rat and the mouse validation data.

In addition to the age association, we also used the mouse data to test the effect of dietary intervention on the rat DNAmAge. Of the n=177 mice, 20 underwent caloric restriction (CR). Using an age-adjusted model, we found that DNAmAge was decreased in CR versus ad libitum fed animals (*Supplementary file 1* Table S3), such that CR was associated with an average decrease of 1.20 months in DNAmAge (s.e.=0.31, p=1.6e-4). Although mice were subjected to CR starting at 14 weeks, the time they remained on the diet varied from about 6.5-23 months, at which point DNAm was assessed. Therefore, we tested whether prolonged CR amplified the deceleration of DNAmAge using a model with an interaction between age and CR. Results showed that DNAmAge was

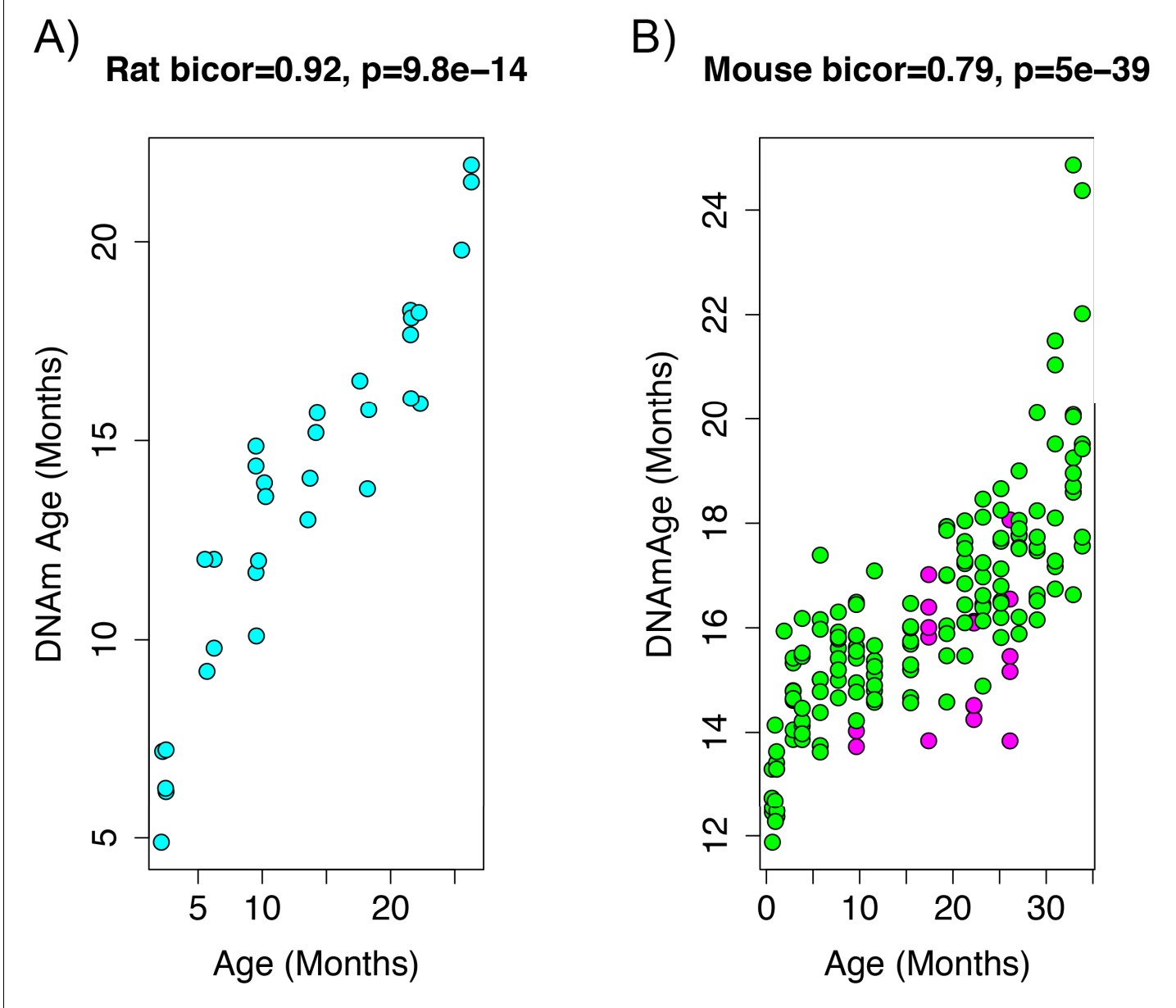

**Figure 2.** Age correlation in rats and mice using the restricted overlapping CpGs. (A) PC1 DNAmAge was re-estimated using only the CpGs that were overlapped between rat and mouse (n = 3,625). Although this variable contained only 0.066% of the CpGs in the original measure, we were able to observe an equivalent age biweight midcorrelation. (B) This variable was then applied to data from C57BL/6 mice and was found to be strongly correlated with age (biweight midcorrelation). We also observed the calorie-restricted mice (magenta) trended towards lower DNAmAge than ad libitum fed animals (green)—significance was tested using OLS regression.

The online version of this article includes the following figure supplement(s) for figure 2:

**Figure supplement 1.** Correlations between duration of caloric restriction and epigenetic age acceleration.

significantly associated with the interaction between chronological age and CR ($\beta$=-0.12, p=9.1e-3). This suggests a negative correlation between CR duration and DNAmAge. To test this directly, we calculated the linear model for DNAmAge regressed on age, using only control animals. We then applied the equation to CR animals to calculate the age residual of DNAM, which we correlated with during of CR. As shown in *Figure 2—figure supplement 1*, duration of CR was correlated with DNAmAge (relative to the expected value based on age at r=-0.55, p=0.012).

RRBS data generated from mouse kidney, kidney-derived induced pluripotent stem cells (iPSCs), lung, and lung-derived iPSCs was used to test whether cellular reprogramming altered DNAmAge. Results (*Figure 3*) showed that the DNAmAge was significantly decreased in both kidney (p=1.13e-4) and lung (p=6.57e-3), by a little over 3 months. Given that the increase in DNAmAge as a function of chronological age is compressed in mice, when refitting so that the slope is equal to 1, the difference between tissue and tissue-derived iPSCs actually translates to ~20 months.

### Deconstruction of epigenetic aging measures

We hypothesize that the DNAmAge measure captures a composite of diverse epigenetic aging phenomena. Thus, in order to deconstruct this aging signal into distinct components from which we can better study mechanisms, we clustered CpGs based on co-methylation patterns across samples. For

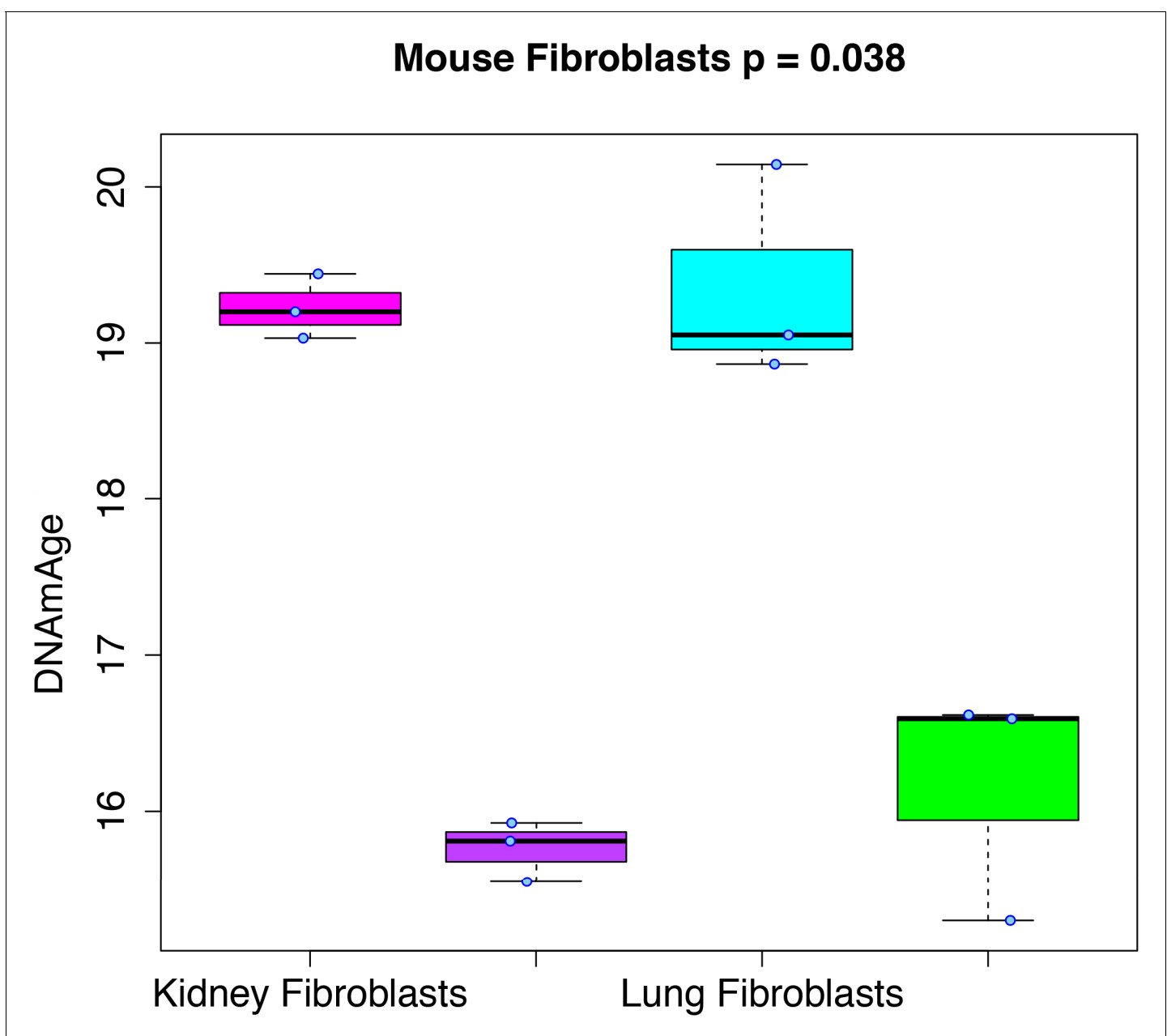

**Figure 3.** Rejuvenation of fibroblast-derived iPSCs. We applied our DNAmAge measure to data from kidney and lung fibroblast controls and derived iPSCs (GSE80672). We find that a significant reduction (Kruskal-Wallis test) in DNAmAge for lung- and kidney-derived iPSCs versus controls.

instance, using the 3,755 CpGs shared across the rat and mouse RRBS data, we performed weighted gene correlation network analysis (WGCNA), with the goal of identifying co-methylation modules. Co-methylation modules represent clusters of tightly related CpGs, whose DNAm values are highly correlated across samples in both the rat and the mouse data. Using a signed network with a power of 1, we identified four co-methylation modules (*Figure 4—figure supplement 1*)—blue (64 CpGs), pink (47 CpGs), purple (44 CpGs), and green (38 CpGs). The majority of CpGs (n=3,432) were not assigned to a module (denoted as the grey module). Next, we tested whether certain modules contributed more to the DNAmAge measure. To do this, we compared the weights of CpGs as a function of which module they were assigned to. We found that the green, purple, and blue modules contained CpGs with higher loadings compared to the grey module. Interestingly, the green module contained CpGs with both the highest positive and lowest negative loadings (CpGs that were strongly hypermethylated and hypomethylated with age, respectively), whereas the purple and blue modules were exclusively made up of CpGs that positively loaded on DNAmAge (CpGs that trended towards hypermethylation with age). Conversely, the pink module was made up of CpGs with loadings at or near 0, suggesting that these CpGs did not contribute much to the overall DNAmAge score (*Figure 4*).

Given the high loadings in the green, blue, and purple modules—suggesting that CpGs in these modules may be driving the original DNAmAge measure—we generate DNAmAge measures based on CpGs within each module (module-specific DNAmAge measures). This was done using the same PCA method in which we constructed the original score (based on all 3625 CpGs), but in this case, it was carried out one-by-one for each module, restricting to CpGs in that module. As seen in *Figure 4—figure supplement 2*, high age correlations were observed for the green module ($r_{rat}$=0.86, $r_{mouse}$=0.58), the purple module ($r_{rat}$=0.88, $r_{mouse}$=0.68), and the grey module ($r_{rat}$=0.89, $r_{mouse}$=0.84), while moderate age correlations were observed for the blue module ($r_{rat}$=0.33, $r_{mouse}$=0.39), and non-significant age correlations were found for the pink module ($r_{rat}$=0.05, $r_{mouse}$=-0.11).

We next evaluated the associations between module-specific DNAmAge (excluding grey) and the age-related physical functioning in rats, CR in mice, and reprogramming in lung and kidney fibroblasts. Results using a fully adjusted linear model (controlling for age and FACS PC1, *Supplementary file 1* Table S3) suggest that the green module DNAmAge (β=0.51, p=4.9e-3) and the blue module DNAmAge (β=0.34, p=3.1e-3) were positively associated with age-related physical functioning, such that higher DNAmAge in these two modules (accounting for chronological age and cell composition) was associated with a more functionally impaired phenotype in rats (*Supplementary file 1* Table S3). When comparing module-specific DNAmAge measures on the basis of CR, we found that all four modules were significantly reduced in CR versus control mice ($p_{blue}$=2.5e-8, $p_{pink}$=2.9e-10, $p_{green}$=5.2e-4, $p_{purple}$=2.2e-3). Finally, only the blue and green DNAmAges showed significant decline after reprogramming in both kidney ($p_{blue}$=5.2e-4, $p_{green}$=2.3e-4) and lung ($p_{blue}$=7.6e-3, $p_{green}$=1.8e-2).

## Genomic features of methylation modules

In order to identify underlying epigenetic mechanisms associated with the four modules, we used the Cistrome Project database (http://cistrome.org) to assess enrichment for binding overlap of transcription factors (TFs), chromatic regulators, histone marks, and variants. The most striking results were for CpGs in the blue module, which exhibited substantial overlap with the repressive histone modification marks H3K9me3 and H3K27me3 and E2F1 TF binding (*Figure 5—figure supplement 1*). The green module also exhibited high enrichment for H3k9me3 and H3k27me3, but to a somewhat lesser degree (*Figure 5—figure supplement 2*). Although other modules showed slight enrichments for TFs, chromatic regulators, and histone marks, they were substantially lower than what was observed in the blue module (*Figure 5—figure supplement 3* and *4*).

Next, we tested for enrichments based on UCSC Genome Browser features. We found that as much as 98% of the CpGs in the blue module and 55% of the CpGs in the green module were located far from TSS ($10^3$ to $10^6$ downstream (*Figure 5A*)). Conversely, only about 10% of CpGs in the overall data (n=3625) were located within this proximity to TSS. Similarly, both the blue and green modules were substantially enriched for CpGs in intergenic regions—nearly every CpG in the blue module is in an intergenic region, as are approximately 60% of CpGs in the green module (*Figure 5B*). This represented enrichments of 7.4-fold and 4.5-fold for the blue and green modules,

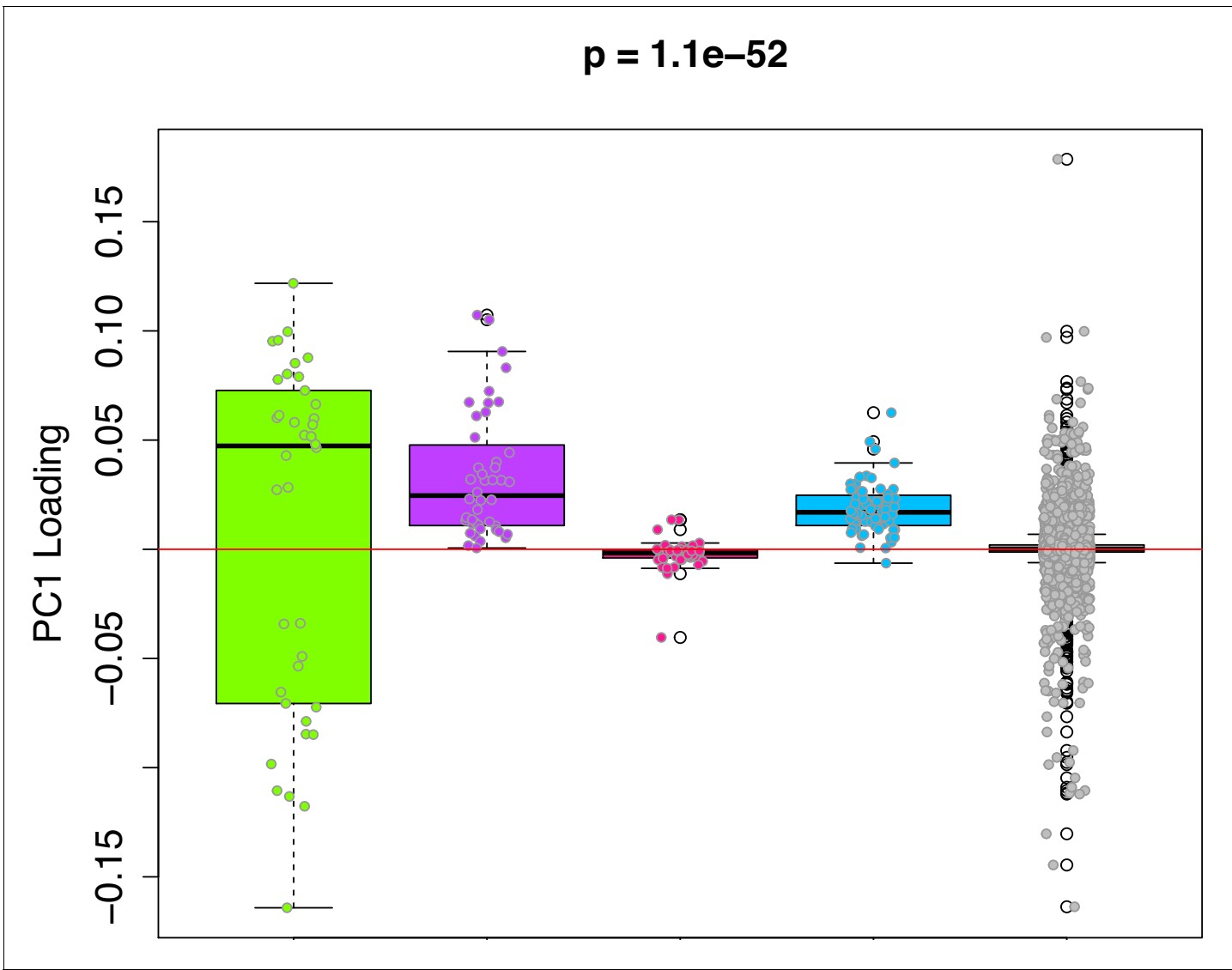

**Figure 4.** PC1 loadings (for overlapped CpGs from mouse data) by module. Each of the 3625 CpGs was assigned to a module (i.e. green, purple, pink, blue, grey). The y-axis shows the loadings from PC1 that was used to signify the DNAmAge measure. The Purple and Blue modules tend to have CpGs with positive loadings, signifying that DNAm levels for CpGs in these modules were more strongly related to higher DNAmAge. The green module had CpGs with both high positive and high negative loadings, suggesting that half the CpGs in this module are hypomethylated in accordance with higher DNAmAge whereas the other half are hypermethylated with higher DNAmAge. p-Values denotes significance using Kruskal-Wallis test.
The online version of this article includes the following figure supplement(s) for figure 4:

**Figure supplement 1.** Dendrogram from WGCNA.

**Figure supplement 2.** Correlations between module-based DNAmAge and age in both rats and mouse.

respectively, relative to all CpGs in the data (only ~13% were found to be located in intergenic regions).

Trends in CpG density was another striking feature that differentiated modules. For instance, results showed that the blue module was highly skewed towards CpGs in CG dense regions (*Figure 5C-D*). The green module showed bimodal distribution, such that about half the CpGs it contains were in highly dense regions and the other half were in very sparse regions. When examining the behaviors of all the CpGs in the models, we found that the CpGs from the green module, in the sparse regions, were those that became hypomethylated with age yet contributed substantially to the overall DNAmAge score—as indicated by their strong negative loadings (*Figure 5D*). In contrast, the green module CpGs in dense regions had high positive loadings, suggesting that

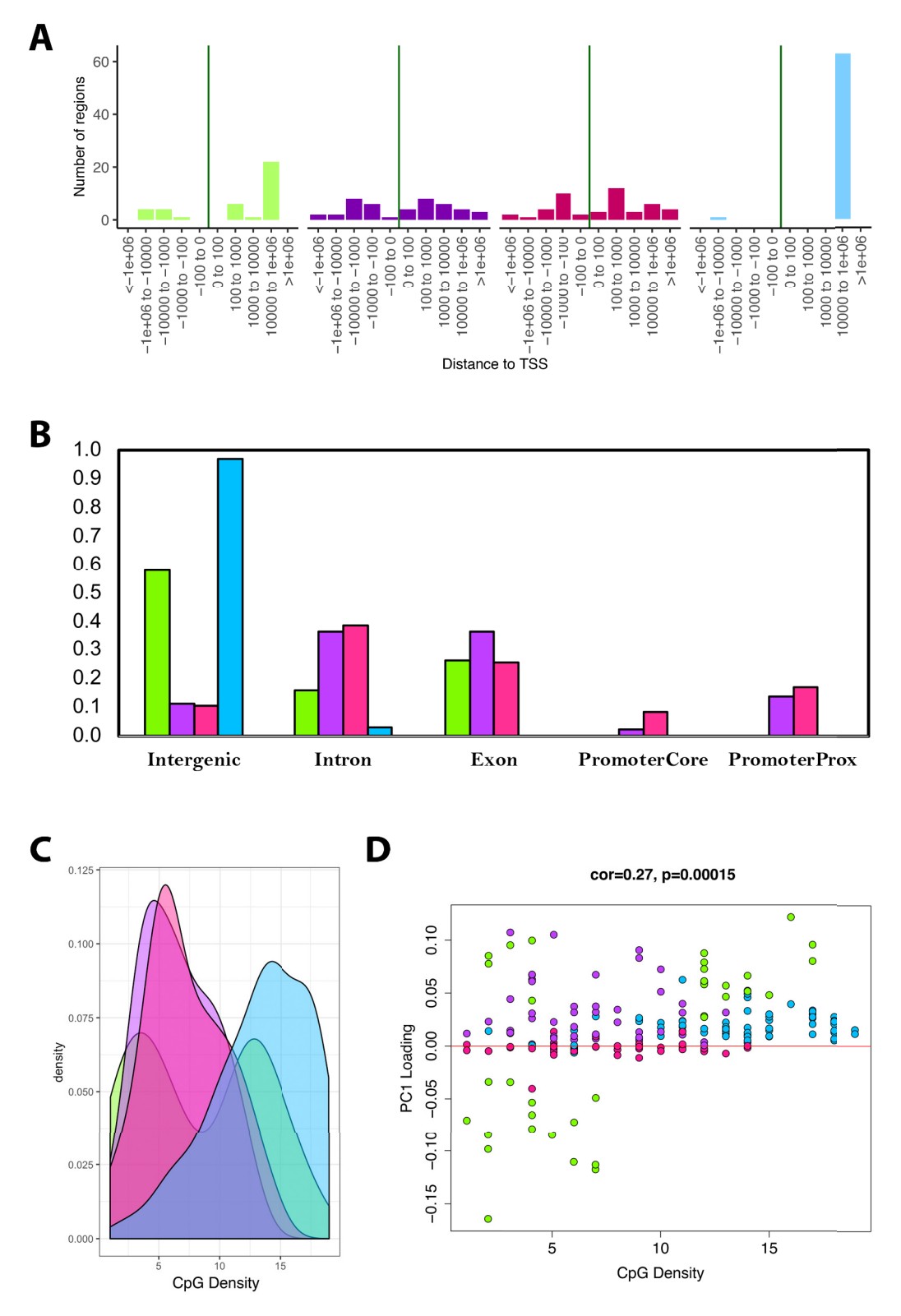

**Figure 5.** Module characteristics. (**A**) Base-pair proximities of CpGs to Transcription Start Sites (TSS), according to module assignment (denoted by color). We find a high proportion of CpGs in the blue module (and to some extent the green module) are located 10,000 to 1,000,000 downstream of TSS. (**B**) Proportion of CpGs in various genomic regions as a function of module (denoted by color). Results suggest that 98% of CpGs in the blue module and 60% of CpGs in the green module are in intergenic regions. (**C**) Distribution of surrounding CpGs densities according to module (denoted

*Figure 5 continued on next page*

*Figure 5 continued*

by color). CpG density was (x-axis) was calculated as the number of CpGs within a 100 bp window (50 bp on either side of the CpG of interest). We observed that the blue module tended to be comprised of CpGs located in regions of higher CpG density (island), while the green module was bimodal, with half the CpGs in it being located in high-density regions, and the other half in low-density regions. (D) We then plotted CpG density as a function of the CpG loading for PC1 that was used to estimate DNAmAge from overlapped CpGs. We find that for the green module, CpGs with strong negative loadings are in low-density regions, while those with high loadings are in both low- and high-density regions.

The online version of this article includes the following figure supplement(s) for figure 5:

**Figure supplement 1.** Transcription factor binding and histone mark enrichment for blue module.
**Figure supplement 2.** Transcription factor binding and histone mark enrichment for green module.
**Figure supplement 3.** Transcription factor binding and histone mark enrichment for purple module.
**Figure supplement 4.** Transcription factor binding and histone mark enrichment for pink module.

hypermethylation of these CpGs was strongly reflective of accelerated aging in the DNAmAge variable. Similarly, the blue and purple modules were made up of CpGs for which hypermethylation was associated with aging, but the difference was that the blue module contained CpGs in dense regions, whereas the purple module contained CpGs in sparser regions. Finally, the pink module contained CpGs that did not appear to have a strong aging signal, nor have defining characteristics in regard to genomic location.

## Discussion

This study presents the first published epigenetic aging clock developed for rats – with special acknowledgement of another manuscript uploaded to biorxiv also demonstrating a rat epigenetic clock (https://www.biorxiv.org/content/10.1101/2020.05.07.082917v1). While it shares many of the same experimental advantages with its close relative, the mouse, the rat represents an exciting rodent model for studying mammalian aging. Its larger size better enables longitudinal tracking and simultaneous assessment of multiple assays and it is also better suited to in vivo brain scanning and performing assessments to study heterogeneity in cognitive aging (*Ellenbroek and Youn, 2016*). In moving forward, our ability to track how molecular features change over time in accordance with functional phenotypes will be critical for disentangling the causal pathways through which aging hallmarks directly contribute to morbidity and mortality. It will also better enable assessments of geroprotective therapeutics given that samples can be taken pre- and post-treatment without causing undue harm to the animal.

Our novel epigenetic clock was shown to strongly correlate with chronological age and relate to physical performance, independent of cell composition and/or chronological age. Furthermore, we show that it is also applicable to mice. For instance, we estimated our epigenetic clock in whole blood samples from C57BL/6 and found that it correlates with age at r=0.79. Moreover, we found that it is decreased in animals that have undergone CR. This is consistent with previous studies showing the CR prevent DNA methylation drift and epigenetic age in mice (*Maegawa et al., 2017*; *Petkovich et al., 2017*). We further observed, that the longer the animals remained on a CR diet, the lower their DNAmAge was relative to their chronological age, suggesting that the longer CR is maintained, the greater the decrease in epigenetic age.

Nevertheless, there were also interesting observations that differentiated the performance of our DNAmAge measure in mice versus rats. Most notably, the functional form of DNAmAge regressed on age. While DNAmAge exhibits a sigmoidal fit with age in rats, this is more dramatic in the mouse data. Furthermore, the measure appears compressed at both younger and older ages, such that the range in mouse chronological age is 0.66-34.4 months (mean=17.0, sd=10.8), yet the range in DNAmAge is 11.1-28.8 months (mean=16.9, sd=2.8). The non-linear fit of DNAmAge as a function of age has been repeatedly reported in both human and rodent samples (*Horvath and Raj, 2018*). One possibility for this observation is that the changes in DNAm reflect other non-linear cellular changes, such as cellular division or mutation accumulation, which increase exponentially in early life and then decelerate upon maturation (*Rozhok and DeGregori, 2016*). Relatedly, the proportion of senescent and/or precancerous cells may accumulate in a non-linear manner with age, such that these changes are slow in early life and then accelerate toward the end of the lifespan (*Zietkiewicz et al., 2009*). The combination of these non-linear patterns in composite DNAmAge measure may therefore

produce the sigmoidal age trajectories that have been observed. When considering the compression of DNAmAge across age in mice versus rats, one explanation is that some of the signals captured in epigenetic clocks are conserved, while other are not. If not all age-related DNAm changes in rats are conserved in mice this could offset, distend, or constrict the DNAmAge measure when applied to mice, even if the overall score still exhibits a significant age correlation.

To understand the 'types' of age-related DNAm changes that comprise the epigenetic clock, we performed weighted network analysis, which enabled us to identify co-methylation modules (*Langfelder and Horvath, 2008*). These modules represent groups of CpGs with highly correlated DNAm levels across samples that are thought to capture specific epigenetic aging phenomena. Five modules were identified, four of which exhibited significant age correlations in both rats and mice. However, our ability to relate the modules to phenotypic data—rather than age alone—allowed us to identify the most promising aging signals. For instance, the two modules with the strongest age correlations (grey and purple) were not related to physical functioning in rats. Conversely, the green and blue modules, which had more conservative age correlations, were related to physical functioning in rats, independent of age. Interestingly, they exhibited deceleration in response to CR and were reset upon reprogramming. The majority of epigenetic clock studies evaluate measures on the basis of their age correlations. Here, we were able to start dissecting signals that are purely age-derived from those that are related to phenotypic changes and, therefore, are more relevant for biological aging. Had we relied exclusively on the robustness of the age predictions, it would have been impossible to pinpoint these two modules as being the most informative.

When examining their characteristics, we found that CpGs in the blue and green modules tended to lie in intergenic regions that were more than 10k bp downstream of TSS. Thus, these signatures are unlikely to represent promoter hypermethylation that directly repress gene transcription. Moreover, enrichment analyses showed that CpGs in these two modules tended to co-locate with H3K9me3, H3K27me3, and E2F1 TF, which suggests they are likely located in constitutive heterochromatin and/or facultative heterochromatin domains (*Saksouk et al., 2015*). Interestingly, these patterns are reminiscent of an epigenetic landscape that has been associated with cellular senescence—another major hallmark of aging. In the paper by *Narita et al., 2003*, the authors described a distinct heterochromatic profile of senescent cells, which is characterized by heterochromatin protein recruitment (including H3K9me), and stable repression of E2F target genes, which have essential roles in cell cycle control. One possible explanation for our findings is that we are capturing the increasing proportion of senescent cells in blood that accompanies aging.

Relatedly, it has been proposed that changes in the distribution of heterochromatin may drive aging. Alterations and/or dysregulation in gene expression that is observed in aging, senescence, and/or cancer may stem from the concurrence of a global loss in constitutive heterochromatin that is accompanied by a redistribution of heterochromatin markers to facultative heterochromatin. This aging signature of heterochromatin redistribution may be influenced to some degree by age-related changes in DNAm stemming from downregulated activity of Dnmt1 (DNAm maintenance during replication) and upregulated activity of Dnmt3a/b (de novo DNAm).

Our results suggest that DNAm changes that may reflect redistribution of heterochromatin show strong age effects in blood from both mouse and rat, may be implicated in the age-associated decline in physical functioning, are slowed in response to CR, and are reset by reprogramming via induction of pluripotency.

Unlike previous DNAm aging signatures, our novel measure was built using an unsupervised approach, which was not directly aimed at chronological age prediction. Interestingly, while we considered the additive effects of multiple PCs in generating the DNAmAge measure, PC1 alone was sufficient to produce a highly age correlated variable in both the training and validation datasets. One explanation is that the F344 rats used in this study were highly homogenous—with the same sex, genetic background, and environment. Therefore, the main variance to be explained was age, which was subsequently captured in PC1. Because the rats may be aging in the same manner without differencing aging patterns/phenotypes, additional PCs were not picking-up age-related changes. That being said, we hypothesize that PCs beyond PC1 will add additional information when assessing aging in heterogenous populations—genetically diverse groups (e.g. humans), environmentally heterogeneous populations, diverse tissues, etc.

While the use of a homogenous group may have facilitated our ability to employ an unsupervised approach to develop an aging measure, it may also limit generalizability. For instance, all animals in

our training and testing sets were male. Therefore, it is unclear whether female rats and/or mice would exhibit a different DNAm pattern with aging. This will be important to determine going forward given the consistent se differences in life expectancy across most species. Another potential limitation was the exclusive use of samples from blood for the majority of our analysis. As a next step, it will be crucial to determine if the signature we identified in our samples is universally exhibited across multiple diverse tissues. Interestingly, most human epigenetic clocks trained solely on DNAm in whole blood show strong age correlations in nearly all non-blood samples—suggesting that the major DNAm changes observed in blood are not blood-specific. As such, we hypothesize the same would be true for our new rodent epigenetic aging measure. In our analysis, we did have some samples that weren't from blood (lung and kidney mouse fibroblasts). For both of these, we did observe an effect of reprogramming, further supporting the idea that the age measure we are capturing is not merely a reflection of changes in blood cells. However, the absolute values of the prediction were not closely aligned with the observed (or expected age) of the cells. Yet, this was also the case for the mouse samples from blood as well.

In moving forward, our ability to continue to unravel the complex signal linking DNAm changes to functional aging outcomes will require experiments in diverse model systems in which epigenetic changes can be both tracked longitudinally and related to other informative molecular and physiological hallmarks of aging.

# Materials and methods

## Key resources table

| Reagent type (species) or resource | Designation | Source or reference | Identifiers | Additional information |
|---|---|---|---|---|
| Strain, strain background (*Rattus norvegicus*, male) | Fischer 344 CDF | NIA Aged Rodent Colony | | |
| Software, algorithm | Fusion 4.0 | Accuscan Instruments, Inc | | |
| Software, algorithm | Rota-Rod Treadmill Control Software | MED-Associates, Inc | SOF-ENV-575 | |
| Commercial assay or kit | QIAamp DNA Blood Mini Kit | Qiagen | 51106 | |
| Commercial assay or kit | Genomic DNA Clean & Concentrator (gDCC) | Zymo | D4010 | |
| Commercial assay or kit | Qubit 1x dsDNA HS Assay Kit | Thermo Fisher Scientific | Q33231 | |
| Antibody | Anti-rat CD3-FITC (Mouse monoclonal) | BioLegend | Cat#: 201403 | FACS (1.0 ul/test) |
| Antibody | Anti-rat CD25-PE (Mouse monoclonal) | BioLegend | Cat#: 202105 | FACS (1.25 ul/test) |
| Antibody | Anti-rat CD8a-PerCP (Mouse monoclonal) | BioLegend | Cat#: 201712 | FACS (5.0 ul/test) |
| Antibody | Anti-rat CD11b/c-PE-Cy7 (Mouse monoclonal) | BioLegend | Cat#: 201818 | FACS (1.25 ul/test) |
| Antibody | Anti-rat CD4-APC-Cy7 (Mouse monoclonal) | BioLegend | Cat#: 201518 | FACS (2.5 ul/test) |
| Antibody | Anti-rat RT1B-AF647 (Mouse monoclonal) | BD Biosciences | Cat#: 562223 | FACS (1.25 ul/test) |
| Antibody | Anti-rat CD45RA-BV421 (Mouse monoclonal) | BD Biosciences | Cat#: 740043 | FACS (1.25 ul/test) |
| Antibody | Anti-rat CD45-BV605 (Mouse monoclonal) | BD Biosciences | Cat#: 740371 | FACS (1.25 ul/test) |

*Continued on next page*

*Continued*

| Reagent type (species) or resource | Designation | Source or reference | Identifiers | Additional information |
|---|---|---|---|---|
| Sequence-based reagent | Immunoprep Reagent System | Beckman Coulter | Cat#: 7546999 | |
| Sequence-based reagent | AMPure XP | Beckman Coulter | CAT#:A63880 | RRBS (variable) |
| Commercial assay or kit | TruSeq Barcoded Adapters | Illumina | CAT#:FC-121-1003 | RRBS |
| Software, algorithm | CutAdapt | | v2.4 | RBBS adapter trimming |
| Software, algorithm | BSSeeker2 | | v2.0.4 | RRBS Alignment / Methylation Calling |

## Animals

All experimental procedures were conducted in accordance with the Guide for the Care and Use of Laboratory Animals and approved by the NIA Animal Care and Use Committee. Male F344 rats were obtained from the NIA Aged Rodent Colony housed at the Charles River Laboratories (Frederick, MD). After receipt into NIA intramural housing facility (Baltimore, MD), animals were housed with Nylabone supplementation and ad libitum access to food (NIH-31 diet) and water. Rats younger than 3 months were housed in groups of three; all other rats were singly housed. Rats were maintained on a 12/12 lighting schedule, with all procedures carried out during the animals' light cycle. Rats were habituated to the facility for at least 3 days before 500 µl of whole blood was collected via retro-orbital bleedings for DNA and FACS analysis. Blood for DNA was collected in heparinized tubes, spun, and the plasma removed; buffy coat and red blood cells were frozen at −80°C until DNA extraction. Blood for FACS analysis was collected in EDTA-treated tubes, chilled on ice, and tested immediately.

The RRBS data from mouse blood and iPSC samples were acquired from Gene Expression Omnibus, under the accession number GSE80672.

## Behavior

Rats were recovered from bleeds for at least eight days prior to behavioral testing. For open field test, rats were placed for 30 min in a 42x42x33cm Plexiglas chamber. Activity was monitored through infrared beams detected with AccuScan Fusion software (Omnitech Electronics; Columbus, OH). On a separate day, rats were tested in a rotarod apparatus (MED-Associates; St Albans, VT) consisting of a 7-cm rotating drum placed 60 cm above a horizontal surface. Rats were placed on the drum rotating at a constant 8 RPM for 5 min; rats that fell off during that time were placed back on. Rats were then tested in three trials with accelerating drum rotation (4-40 RPM) and tested for latency to fall up to 5-min maximum trial duration. Rats recovered in their homecage for 15 min between trials. Nails of hindlimbs were trimmed in old rats (18+ mo-old) 1 day prior to testing.

## FACS

One hundred µL of whole rat blood from chilled EDTA-treated tubes was stained and then processed using a Beckman Coulter TQ-prep and the Beckman Coulter immunoprep reagent system. Immunophenotyping data was acquired on a BD FACSCanto II and analyzed using BD FACSDiva. Antibodies used for fluorescence analysis are as follows: FITC-conjugated anti-rat CD3 (clone 1F4), PE-conjugated anti-rat CD25 (clone OX-39), PerCP-conjugated anti-rat CD8a (clone OX-8), PE-Cy7 conjugated anti-rat CD11b/c (clone OX-42), APC-Cy7 conjugated anti-rat CD4 (clone W3/25) from Biolegend (City, state), and AF647-conjugated anti-rat RT1B (clone OX-6), BV421-conjugated anti-rat CD45RA (clone OX-33), and BV605-conjugated anti-rat CD45 (clone OX-1) from BD Biosciences (City, state).

## DNA analysis, RRBS libraries, sequencing alignment, and methylation matrix assembly

Following proteinase K and RNAse A treatments, DNA was isolated from rat blood cells using QIAmp DNA Mini Kit (Qiagen, City, state) following manufacturer's instructions using a QIAcube automated device. DNA was eluted from columns in 200 µL of AE buffer, concentrated in 20 µL 10 mM Tris-HCl, pH 8.5, 0.1 mM EDTA using Genomic DNA Clean & Concentrator-10 (Zymo), City, state, and quantified using a Qubit 2.0 (Life Technologies, City, state). RRBS libraries were generated as described previously (*Orozco et al., 2015*; *Smith et al., 2009*). Briefly, 100 ng of isolated DNA was digested with MspI restriction enzyme (NEB, Ipswich, MA), carried out end-repair/adenylation (NEB) and ligation with TruSeq barcoded adapters (Illumina, San Diego, CA). DNA fragment length between 200 and 300bp were selected with SPRI magnetic beads (ABM, Richmond, BC, Canada), followed by bisulfite treatment (Millipore, Billerica, MA), and PCR amplification (Bioline, Taunton, MA). The libraries were sequenced by multiplexing 8 libraries per lane on the Illumina HiSeq2500 sequencer, with 100 bp single-end reads.

Sequencing reads were trimmed using CutAdapt (*Martin, 2011*) to remove adapter sequences. Trimmed reads were mapped and methylation beta values called from the alignment file using BS-Seeker2 (*Guo et al., 2013*). We filtered aligned data to keep only cytosines with greater than 10x coverage in at least 80% of the samples. Methylation values for samples at sites with lower than 10x coverage were set to null. Missing values were imputed using a kNN sliding window; missing methylation values were assigned the average value of the five nearest neighbors by Euclidean distance within a 3Mb window.

## Statistical analysis

Elastic net penalized regression was used to generate a DNAm predictor of chronological age in rats based on RRBS data containing approximately 5.5 million cytosines. Prior to training, rats were grouped into training and testing sample based on age. For instance, rats ages 2m, 6m, 10m, 14m, 18m, 22m, and 26m were selected for the test set (n=32), while all others were included in the training set (n=104). Next PCA was run using the rats in the training set and elastic net was used to train a predictor of age based on PCs rather than individual CpGs. Given that the 10-fold cross-validation for the elastic net model showed PC1 alone produced the best predictor of age (*Figure 1—figure supplement 1*), PC1 was used as an estimate of DNAmAge. Thus, this measure contained the information from all CpGs (weighted according to contribution/loadings on PC1). The eigenvalues for PC1 were converted to units of months by regression chronological age on PC1 eigenvalues in the training sample. The PC-based DNAm was the product of eigenvalues and the beta coefficient from the linear regression, with the addition of the constant (intercept term) from the regression model. Both the PC based and the CpG based DNAmAge measures were estimated in the test samples by applying the equations from the training data. They were then evaluated based on biweight midcorrelations with age and multivariate linear regression analysis to assess the association between them and PC1 from a PCA of phenotypic variables (i.e. rotarod and open filed), after adjusting for age and cell composition (PC1-5 from PCA of FACS data).

LiftOver chain file from rn6 to mm10 genome assembly was downloaded from UCSC genome browser (https://genome.ucsc.edu/index.html) to provide the alignment from rat to mouse genome. rtracklayer liftOver function (https://www.bioconductor.org/help/workflows/liftOver/) was used to load the chain file and map the rat CpG to corresponding mouse genome coordinates. We then assessed CpG overlap between the mouse and rat data and found that 3625 of the 5.5 million CpGs were available in all datasets. As a result, we reran PCA in the training rat sample and fit a new DNAmAge measure based on the PCs from the 3625 CpGs. We also re-evaluated age correlations and associations with phenotypic variables in the rat test data. We then fit this DNAmAge predictor in the mouse data and used biweight midcorrelations to assess age associations, and multivariate linear regression analysis to assess the effect of CR on DNAmAge, relative to chronological age, as well as the effect of iPSC reprogramming in mouse lung and kidney fibroblasts.

To identify co-methylation modules, we applied consensus WGCNA to the mouse and rat RRBS data. For this analysis, we input data as a 'signed' topological overlap matrix, with a power=1. We also used deepSplit=1, minModuleSize=25, and minKMEtoStay=0.4. After grouping CpGs in

modules, we generated module-specific DNAmAge measures based on PCA. We then tested for age association and phenotype associations in a manner consistent with models we have previously run.

Finally, LOLA (http://databio.org/regiondb) and Cistrome (http://cistrome.org) were used to assess enrichment for binding overlap of transcription factors, chromatic regulators, histone marks, variants, and genomic locations. For enrichment we used a background comprising all 3625 CpGs from the overlapped mouse and rat RRBS data. Enrichment was conducted independently for each of the four modules (excluding grey). Mouse (mm10) region sets were used for all enrichment analysis, given that Cistrome and LOLA only contain databases for human and mouse.

## Acknowledgements

We acknowledge Quia Claybourne for technical assistance with rat behavior tests. This work was supported by funding from the Intramural Research Program of the National Institute on Aging/NIH. As well as NIH/NIA Grant 5R00AG052604.

## Additional information

### Funding

| Funder | Grant reference number | Author |
| --- | --- | --- |
| National Institute on Aging | 1 ZIA AG000993-01 | Rafael de Cabo Luigi Ferrucci |
| National Institute on Aging | 5R00AG052604-04 | Morgan Levine |
| National Institute on Aging | P30AG021342 | Morgan Levine |

The funders had no role in study design, data collection and interpretation, or the decision to submit the work for publication.

### Author contributions

Morgan Levine, Conceptualization, Formal analysis, Supervision, Validation, Investigation, Visualization, Methodology, Writing - original draft, Writing - review and editing; Ross A McDevitt, Conceptualization, Data curation, Formal analysis, Investigation, Methodology, Writing - review and editing; Margarita Meer, Colin Farrell, Data curation, Formal analysis, Writing - review and editing; Kathy Perdue, Andrea Di Francesco, Theresa Meade, Conceptualization, Data curation, Methodology, Writing - review and editing; Kyra Thrush, Meng Wang, Formal analysis, Writing - review and editing; Christopher Dunn, Data curation, Formal analysis, Methodology; Matteo Pellegrini, Data curation, Supervision; Rafael de Cabo, Conceptualization, Data curation, Supervision, Funding acquisition, Project administration, Writing - review and editing; Luigi Ferrucci, Conceptualization, Data curation, Supervision, Funding acquisition, Methodology, Writing - review and editing

### Author ORCIDs

Morgan Levine https://orcid.org/0000-0001-9890-9324
Ross A McDevitt https://orcid.org/0000-0003-3722-9047
Margarita Meer http://orcid.org/0000-0001-8249-7097
Andrea Di Francesco https://orcid.org/0000-0001-6867-8203
Colin Farrell https://orcid.org/0000-0002-3138-6108
Christopher Dunn https://orcid.org/0000-0001-7899-0110
Matteo Pellegrini https://orcid.org/0000-0001-9355-9564
Rafael de Cabo https://orcid.org/0000-0003-2830-5693
Luigi Ferrucci https://orcid.org/0000-0002-6273-1613

### Ethics

Animal experimentation: All animal protocols were approved by the Animal Care and Use Committee (277-TGB-2016) of the National Institute on Aging.

Decision letter and Author response
Decision letter https://doi.org/10.7554/eLife.59201.sa1
Author response https://doi.org/10.7554/eLife.59201.sa2

# Additional files

## Supplementary files

• Source data 1. Rat estimated DNAmAge and traits. DNAmAge is estimated using the PCA-based method with input from the 3625 overlapped CpGs.

• Source data 2. Mouse estimated DNAmAge and traits. DNAmAge is estimated using the PCA-based method with input from the 3625 overlapped CpGs.

• Supplementary file 1. Table S1. PC1 loading for phenotypic variables. Table S2. Phenotypic associations with DNAmAge based on overlapped CpGs. Table S3. Association between DNAmAge and caloric restriction (CR) in C57BL/6. Table S4. Phenotypic associations with DNAmAge based on overlapped CpGs. Table S5. Association between DNAmAge and individual phenotype variables PC1 loading for phenotypic variables.

• Transparent reporting form

## Data availability

DNA methylation data is available via Gene Expression Omnibus (GEO) accession GSE161141.

The following dataset was generated:

| Author(s) | Year | Dataset title | Dataset URL | Database and Identifier |
|---|---|---|---|---|
| Levine M, de Cabo R, Ferrucci L | 2020 | A rat epigenetic clock recapitulates phenotypic aging and co-localizes with heterochromatin-associated histone modifications | https://www.ncbi.nlm.nih.gov/geo/query/acc.cgi?acc=GSE161141 | NCBI Gene Expression Omnibus, GSE161141 |

The following previously published dataset was used:

| Author(s) | Year | Dataset title | Dataset URL | Database and Identifier |
|---|---|---|---|---|
| Petkovich DA, Podolskiy DI, Lobanov AV, Gladyshev VN | 2017 | Using DNA methylation profiling to evaluate biological age and longevity interventions | https://www.ncbi.nlm.nih.gov/geo/query/acc.cgi?acc=GSE80672 | NCBI Gene Expression Omnibus, GSE80672 |

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
