## [Decision Letter]

**Acceptance summary:**

Levine et al. present a DNA methylation-based clock trained on rat chronological age, using longitudinal whole blood samples from a cohort of male rats. This represents the first clock for predicting rat aging, which is an important advance in the aging research field as rats offer important practical advantages over mice, particularly the ability to draw recurring sizable blood samples without harm to the animal. This longitudinal sampling, combined with the proposed model, will enable longitudinal studies of aging biology and potential evaluation of interventions.

**Decision letter after peer review:**

Thank you for submitting your article "A rat epigenetic clock recapitulates phenotypic aging and co-localizes with heterochromatin" for consideration by *eLife*. Your article has been reviewed by two peer reviewers, and the evaluation has been overseen by a Reviewing Editor and Jessica Tyler as the Senior Editor. The following individuals involved in review of your submission have agreed to reveal their identity: Peter Laird (Reviewer #1); Bjorn Schumacher (Reviewer #2).

The reviewers have discussed the reviews with one another and the Reviewing Editor has drafted this decision to help you prepare a revised submission.

Summary:

The manuscript by Levine et al. presents a DNA methylation-based clock trained on rat chronological age, using longitudinal whole blood samples from a cohort of male F344 rats. This represents the first epigenetic clock for predicting rat aging, and thus a significant extension of what have been defined in mice and humans. In particular, rat models provide a number of advantages over mouse models especially with respect to the possible recurring measurement of blood profiles without harm to the animals. The authors show that the rat DNAmAge derived from the longitudinal sampling and a Principal Component-based modeling also correlates with phenotypic aging in mice and rejuvenation-related phenotypes conferred by an age intervention such as calorie restriction in mice as well as in fibroblast-derived iPSCs. This longitudinal study design, combined with the proposed model, is powerful with the demonstrated potential to evaluate the effects of aging interventions. Thus this study will certainly be of broad interest to the community of researchers studying the biology of aging. However, as detailed below, a number of concerns were also raised, relating in large part to insufficient clarity in the current version with respect to the authors' presentation of methods and data.

Essential revisions:

1) Abstract:

Is the number behind physical functioning a p-value?

2) Subsection “Age differences in DNAm”

i) There are 27 age groups; they chose 32 animals for the test set, the exact age distribution should be indicated.

ii) They should mention the panel number in the text, not just Figure 1.

iii) The order of the panels is different from the appearance in the text. Reorder.

iv) Paragraph three, a plot for the first principle components, maybe also color coded for age, should be included so that the reader can see it for themselves.

v) The statement “PC1 captures 6.7 % of variance” is not directly clear from the plot and should instead to visible in Figure 1A. On the plot it looks like PC1 is at ~0.48. Is the y-axis here in percent? Also the axis in the plot should be changed so that the axis spans the whole data.

vi) The authors mention that PC1 explains 87 % of the variance in age, but it is not clear how this was computed. It also was not really clear how they computed this (Figure 1C). Clarify.

vii) They used regression to get the relationship between the PC1 and age. But then how did they use this to compute the age in the test sample? The authors should explain this further.

viii) In the figure legend they write: “multiplying the coefficient by PC1 and adding the constant in the test sample”. Do they mean they multiply the coefficients of the regression line by PC1? Which constant do they mean and how and what exactly do they add?

3) Materials and methods:

i) It is not clear what the text "Next PCA was run using the rats in the training set and elastic net was used to train and predictor of age based on PCs rather than individual CpGs" means. It sounds like the authors used all PCs instead of just PC1. Clarify.

ii) They mention that including more PCs into the predictor does not improve the prediction. Show a plot or table for this claim.

iii) For the DNAm associations with FACS and Phenotypic variables:

a) Table S1 in Supplementary file 1: Explain what exactly the β coefficient is (the degree of change in the outcome variable, i.e. DNAmAge, for every unit of change in the predictor variable, i.e. Age, or Pheno PC1,.)

b) Why is the β coefficient between phenotypic PC1 and DNAmAge increasing, when taking more PCs of the FACS PCA into account? Shouldn't it decrease? Clarify.

c) In Figure 1—figure supplement 2 and Table S1 in Supplementary file 1, include the original data.

4) Validation in C57BL/6 Mice:

i) A correlation line in Figure 2 (as they did before) should be added.

ii) Figure 2—figure supplement 1: Verify whether the y-axis label is correct. It might be the difference between the true and the DNAmAge, and not the DNAmAge itself.

5) Deconstruction of epigenetic aging measures:

i) It would be interesting to do the same analysis for the whole rat dataset, instead of the smaller rat-mice dataset.

ii) Why did they choose a power of 1? Usually the smallest power is chosen so that the network is scale-free.

iii) In the text they write Figure S3, but mean S4.

iv) The order of the text and the panels in the figure is not the same.

v) Should the axis be scaled the same?

vi) The correlation coefficients for the pink module in the text and the plot are different

vii) They write “using a fully adjusted linear model ([…] Table S2)”, but Table S2 is not for the specific modules.

viii) They reference the table again, maybe they forgot to include the right table? otherwise this is confusing.

6) Genomic Features of Methylation Modules:

i) Include a figure for the overlaps.

ii) What supplemental material?

7) Discussion:

i) Paragraph one: Horvath's recent paper: Reversing age: dual species measurement of epigenetic age with a single clock (biorxiv 08.05.2020) reports an epigenetic aging clock in rats and should be cited here accordingly.

ii) Paragraph six: “using PCA, rather than elastic net”: in the Materials and methods they write: “PCA was run using the rats in the training set and elastic net was used to train and predictor of age based on PCs”, so it is a combination of both.

---

## [Author Response]

Essential revisions:1) Abstract:Is the number behind physical functioning a p-value?

Yes. Fixed to reflect this.

2) Subsection “Age differences in DNAm”i) There are 27 age groups; they chose 32 animals for the test set, the exact age distribution should be indicated.

We have moved the following statement from Materials and methods to Results:

“For instance, rats ages 2m, 6m, 10m, 14m, 18m, 22m, and 26m were selected for the test set (n=32), while all others were included in the training set (1m, 3m, 4m, 5m, 7m, 8m, 11m, 12m, 13m, 15m, 16m, 17m, 19m, 20m, 21m, 23m, 24m, 25m, 27m).”

ii) They should mention the panel number in the text, not just Figure 1.iii) The order of the panels is different from the appearance in the text. Reorder.

Done

iv) Paragraph three, a plot for the first principle components, maybe also color coded for age, should be included so that the reader can see it for themselves.

Figure 1C plots PC1 against age. We have attempted to make this clearer in the text.

v) The statement “PC1 captures 6.7 % of variance” is not directly clear from the plot and should instead to visible in Figure 1A. On the plot it looks like PC1 is at ~0.48. Is the y-axis here in percent? Also the axis in the plot should be changed so that the axis spans the whole data.

We apologize, this was a typo and should have been 4.7% rather than the 6.7% that was stated.

vi) The authors mention that PC1 explains 87 % of the variance in age, but it is not clear how this was computed. It also was not really clear how they computed this (Figure 1C). Clarify.

Variance explained in age by PC1 can be computed by squaring the correlation coefficient. As shown in Figure 1 that value is r=0.93 (thus r2=0.87). However, because this may be confusing to readers, we now report it as a correlation coefficient rather than variance explained. Example: “PC1 was strongly correlated with age (r=0.93) in our independent test sample (Figure 1C)”

vii) They used regression to get the relationship between the PC1 and age. But then how did they use this to compute the age in the test sample? The authors should explain this further.

We only used the regression to convert the standardized units for PC1 eigenvalues to units of months. We have added the following to the text.

“Because eigenvalues for PC1 are in standardized units, we converted these to units of months to generate an interpretable estimate of DNAmAge. This was done using the β coefficient and intercept values from a linear regression of chronological age regressed on PC1. To convert to units of months, PC1 eigenvalues were multiplied by the β coefficient (0.405) and summed with the intercept value (13.966). This measure of DNAmAge is perfectly correlated (r=1) with the original PC1 eigenvalues, thus, this rescaling does not alter our results. Rescaling using the coefficient 0.405 and constant 13.966 was applied or all subsequent validation in independent test samples.”

viii) In the figure legend they write: “multiplying the coefficient by PC1 and adding the constant in the test sample”. Do they mean they multiply the coefficients of the regression line by PC1? Which constant do they mean and how and what exactly do they add?

See response to comment #7. We have now added several lines to the main text to describe this procedure.

3) Materials and methods:i) It is not clear what the text "Next PCA was run using the rats in the training set and elastic net was used to train and predictor of age based on PCs rather than individual CpGs" means. It sounds like the authors used all PCs instead of just PC1. Clarify.

We agree this was not clear. We have added the following:

“Next PCA was run using the rats in the training set and elastic net was used to train a predictor of age based on PCs rather than individual CpGs. Given that the 10-fold cross-validation for the elastic net model showed PC1 alone produced the best predictor of age (Figure 1—figure supplement 1), PC1 was used as an estimate of DNAmAge. Thus, this measure contained the information from all CpGs (weighted according to contribution/loadings on PC1). (…) They were then evaluated based on biweight midcorrelations with age and multivariate linear regression analysis to assess the association between them and PC1 from a PCA of phenotypic variables (i.e. rotarod and open filed), after adjusting for age and cell composition (PC1-5 from PCA of FACS data).”

ii) They mention that including more PCs into the predictor does not improve the prediction. Show a plot or table for this claim.

Figure 1—figure supplement 1 has been added. It shows that based on 10-fold cross-validation, the elastic net model with the lowest MSE was that with only one variable (which happened to be PC1).

iii) For the DNAm associations with FACS and Phenotypic variables:a) Table S1 in Supplementary file 1: Explain what exactly the β coefficient is (the degree of change in the outcome variable, i.e. DNAmAge, for every unit of change in the predictor variable, i.e. Age, or Pheno PC1,.)

PCs for Phenotypic variables and FACS variables were standardized so that they have a standard deviation of 1. Given that the DNAmAge variable is in units of (months), the β coefficients represent the increase in DNAmAge (in units of months) associated with every 1 s.d. increase in either PC1 FACS or PC1 Pheno. This has been clarified as a footnote.

b) Why is the β coefficient between phenotypic PC1 and DNAmAge increasing, when taking more PCs of the FACS PCA into account? Shouldn't it decrease? Clarify.

It is not increasing. It is decreasing from 1.46 to 1.29, as shown in Author response table 1 and as illustrated from our text.

“In Model 3, we included PC1 from the FACS data (standardized to have s.d.=1) as an additional covariate. In doing so, we observed further attenuation of the age association and a small attenuation of the phenotypic association, suggesting that cell composition may account for a small proportion of the associations between DNAmAge and either age and/or physical functioning.”

**Author response table 1. resptable1:** 

	Β Coefficient (P-value)			
	Model 1	Model 2	Model 3	
DNAmAge (PC1)				
	Age	0.75 (6.0e-15)	0.63 (1.27e-11)	0.42 (4.4e-4)
	Pheno PC1	--	1.46 (3.4e-3)	1.29 (5.9e-3)
	FACS PC1		--	1.91 (2.7e-2)

Pheno PC1 and FACS PC1 are standardized variables, such that they have a standard deviation of 1. Given that the DNAmAge variable is in units of (months), the β coefficients represent the increase in DNAmAge (in units of months) associated with every 1 s.d. increase in either PC1 FACS or PC1 Pheno.

c) In Figure 1—figure supplement 2 and Table S1, include the original data.

The original data has been added in Figure 5—figure supplement 3 and 4, and Table S4 in Supplementary file 1. Overall, we find strong age associations for nearly all open field and rotarod variables—except for time in center and time in stereotypy (Figure S10). The same is true in the multivariate model, such that DNAmAge is also significantly associated with the phenotype variables after adjusting for age and FACS (Table S4). When looking at age associations with individual FACS variables, we find strong positive and negative correlations. For instance, age has a strong negative association with Memory CD8^+^ Cytotoxic T cells and B cells; and a strong positive association with total and percent myeloid cells and activated T cells.

4) Validation in C57BL/6 Mice:i) A correlation line in Figure 2 (as they did before) should be added.

Done

ii) Figure 2—figure supplement 1: Verify whether the y-axis label is correct. It might be the difference between the true and the DNAmAge, and not the DNAmAge itself.

The y-axis label “DNAmAge (Adj. Age)” is correct. This signifies the residual when regressing DNAmAge on chronological age. It represents how much epigenetically older/younger a mouse is compared to what is expected for its chronological age. We use this value rather than the δ (DNAmAge-Age) because the fit with age is non-linear, thus the δ produces biased results (with chronologically older samples being underestimated and chronologically younger samples being over estimated). This is something that has been consistantly reported in the literature (especially related to human epigenetic clocks), which is why the residual (often called age acceleration) has been used in most papers—as we are doing here.

Nevertheless, we have added text to better clarify what the y-axis represents in the legend.

5) Deconstruction of epigenetic aging measures:i) It would be interesting to do the same analysis for the whole rat dataset, instead of the smaller rat-mice dataset.

Generally, because it is computationally intensive, WGCNA cannot be applied to extremely large datasets. Even with ~5k variables it is recommended that WGCNA be run in blocks, in which variables are split and then remerged. However, with 5.5 million variables WGCNA is not computationally feasible (See: https://peterlangfelder.com/2018/11/25/blockwise-network-analysis-of-large-data/). That being said, we hypothesize that similar results would come out of the data given that the methylome is extremely redundant and that we were able to capture almost the identical age signal even when restricting the data to overlapped CpGs. Given that this is an important point, we have made sure to acknowledge it in the Discussion section.

ii) Why did they choose a power of 1? Usually the smallest power is chosen so that the network is scale-free.

The reviewer is correct. WGCNA has traditionally been applied to RNA-seq or similar data, and in such instances the goal is to generate networks that fit scale-free topology. However, theoretically, we don’t feel this applies to the current application as there is no evidence that clustering of DNAm patterns should be scale-free similar to something like a protein-protein interaction network or gene expression pathway. Additionally, the way in which we are thinking of networks based on DNAm data is that they represent informational networks (akin to an operating system of the cell), whose emergent properties are exhibited in different cellular and/or physiological phenotypes. Additionally, it was recently reported that most networks, including informational ones are not scale-free, and that scale-free topology in complex systems is actually quite rare (https://www.nature.com/articles/s41467-019-08746-5). As such, we took a conservative approach in that we did not adjust adjacency measures to force a scale-free topology on the network.

iii) In the text they write Figure S3, but mean S4.iv) The order of the text and the panels in the figure is not the same.v) Should the axis be scaled the same?

Fixed

vi) The correlation coefficients for the pink module in the text and the plot are different.

They are the same (just rounded to the second decimal)

vii) They write “using a fully adjusted linear model ([…] Table S2)”, but Table S2 is not for the specific modules.

We apologize for this oversight and have added the correct table (Table S3 in Supplementary file 1).

viii) They reference the table again, maybe they forgot to include the right table? otherwise this is confusing.

Fixed per response to comment #vii

6) Genomic Features of Methylation Modules:i) Include a figure for the overlaps.

Added

ii) What supplemental material?

Fixed—we meant to include these figures.

7) Discussion:i) Paragraph one: Horvath's recent paper: Reversing age: dual species measurement of epigenetic age with a single clock (biorxiv 08.05.2020) reports an epigenetic aging clock in rats and should be cited here accordingly.

We apologize for this. The paper by Horvath et al. was uploaded to biorxiv after we had started submitting this paper. The citation has been added.

ii) Paragraph six: “using PCA, rather than elastic net”: in the Materials and methods they write: “PCA was run using the rats in the training set and elastic net was used to train and predictor of age based on PCs”, so it is a combination of both.

We have attempted to clarify this. Elastic net suggested that PC1 alone sufficiently captures the age signal. Therefore, the resulting measure is solely based on PCA.